# Green-Roof: The Role of Community in the Substitution of Green-Space toward Sustainable Development

**Sri Yuliani** [1,*] **, Gagoek Hardiman** [2] **and Erni Setyowati** [2]

1   Architecture Department, Universitas Sebelas Maret, 57126 Surakarta, Indonesia
2   Faculty of Engineering, Universitas Diponegoro, 50275 Semarang, Indonesia; ggkhar@yahoo.de (G.H.); ernisetyowati@arsitektur.undip.ac.id (E.S.)
*   Correspondence: sriyuliani71@staff.uns.ac.id; Tel.: +62-8139-235-6722

**Abstract:** The research challenge in the problem of the sustainable development goals is to find solutions for development control through the application of green roofs in residential areas, which is the feasibility of the role of the community. This research was based on the discipline of architecture by considering the role of the community in managing architectural green-space substitution. The purpose of this study was to identify patterns of the role of the community for green roofs feasibility based on housing, conducted in some parts of Sumatra, Java, and East Nusa Tenggara, Indonesia. Data were collected using a survey questionnaire. Data were calculated based on percentages and analyzed using the chi-square method. The results indicated that the optimization of the community role was needed for the sustainability of the green-roof from all economic levels, both urban and rural. The aspect of public awareness and knowledge of the benefits of the green-roof is very high, but the object of the green-roof in Indonesia is still very limited because participation is still not optimal. Therefore, it is necessary to promote the green-roof application to the community by adapting local culture in sustainable green-roof technology innovation.

**Keywords:** green-roof; community role; substitution of green-space; sustainable development

## 1. Introduction

The basic consideration that forms the background of this green-roof research in the field of architecture is the publication of the declaration of the sustainable development goals, specifically related to the development control for environmental conservation and the fulfillment of food necessities. The development of spatial expansion in buildings was not only developed for the purpose of building appearance but has also accommodated the efforts in dealing with environmental problems. Expansion of space in buildings in the field of architecture is so rapid that it reaches the expansion of the roof, that is, in the form of the green-roof.

### 1.1. Prospect of Green Roofs toward Sustainability Development

The benefits of green-roof contribute to some environmental aspects, including increasing biodiversity, reducing the effects of urban heat island [1–5], decreasing the thermal temperature of buildings [6–8]; on the economic aspect, including reducing maintenance costs and lowering operational costs [9–11]; and on aspects of social space, where people have access to green areas [12–14]. Economically, green-roof initiatives are feasible under certain conditions, but consideration needs to be paid to the construction costs of green-roof for the entire roofs of the buildings [9]. Another benefit, added by Muhammad Shafique et al. (2018), is through research on green-roof related to hydrology in

Seoul. The application of large green-roof provides promising results for rainwater management in urban residential areas [5,15–19]. The environmental benefits, also added by Takanori Kuronuma et al. (2018), stated that the green-roof installed in buildings broadly contributes to $CO_2$ reduction [20,21].

In another study, Emrah Yalcinalp et al. (2017) considered the use of vegetation on green-roofs [22]. The green-roof is a method of sustainable residential roofing for urban areas, taking into account that roof vegetation will support the ecology of urban areas. In this research, the criterion for local vegetation is the right choice as part of the green-roof for urban dwelling. Variants of green-roof vegetation found in the research form the basis for sustainable green-roof research recommendations. The criteria for natural roof vegetation must be chosen from plants that adapt to the roof area and climate, in that it can be arranged in harmony with the ecological conditions on the roof of the house. Other results show that practical problems other than vegetation types, i.e., economic and environmental differences, have a large impact on the green-roof preferences [10]. The green-roof sustainability is expected to be able to turn green the urban areas in that it will become more important in overcoming environmental challenges. Additionally, Rachel Gioannini et al. (2018) stated that the use of local plant species would have the prospect of sustainability compared to non-local plants [23]. In view of regional criteria, the application of the green-roof has different specifications according to climate characteristics [24]. Differences are not limited to technology or plant variations, but also the appreciation and knowledge of the local community who contribute to the sustainability of the green-roof. Areas with tropical climates require different management concepts from sub-tropical, dry, or other areas, whether in urban or rural areas. The interaction between the urban population and the green environment will be stronger, in that this study recommended that the relevance of the green-roof study needs to be improved.

Some European and American developed countries have turned to green-roof as the main concept in sustainable development [25]; even the construction industry is focused on the sustainability of environmental-friendly building practices. Green-roof sustainability is determined by several development actors, including building owners, government, and industry [13,25,26], as described in Figure 1a. The green-roof research is related to energy efficiency efforts. However, a study on the sustainability of energy efficiency, with a focus on the role of the community, by Zhang et al. (2018) stated that the role of the people who inhabit permanently in buildings was crucial for energy use [27]. The role of the community in each dwelling is of concern to be investigated because it has a greater potential for sustainability compared to people who live together in buildings. Studies indicate that people's perceptions of interpreting space can be different from the purpose of the space provided by the government through the design of the planner [28]. In addition, community role research conducted by Sri Yuliani et al. (2018) also concluded that one of the determinants of sustainable development was community participation in active roles accommodated in the design [29], as described in Figure 1b. Based on the recommendations of this study, in terms of assessing the prospect of green-roof sustainability, it could target the community's role in housing.

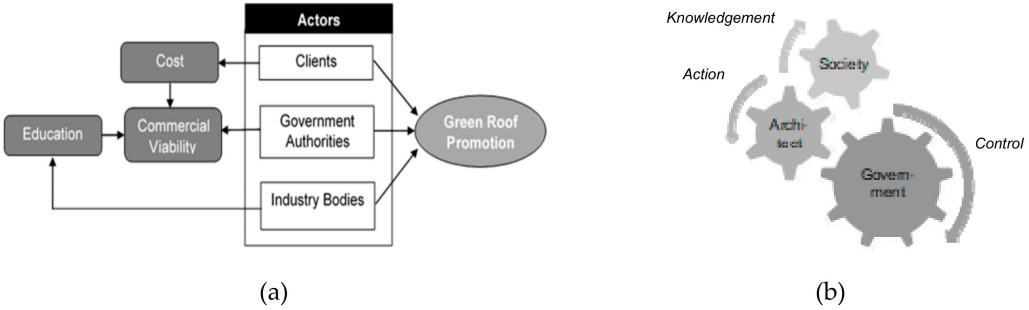

(a)　　　　　　　　　　　　　　　　　　　　　　(b)

**Figure 1.** The network of interactions among key factors, affecting the promotion of green-roof: (**a**) model of the network by Nicole Tassicker et al.; (**b**) network model by Sri Yuliani et al. [29].

Nicole Tassicker et al. (2016) further added that although the country is not yet fully aware of the potential of the green-roof industry, Australia is considered to have promising potential for the future; there must be legislative changes to support it or greater education in the industry [25]. Government authorities need to adjust policy settings to better encourage the use of green roofs, while the industries are required to organize better and more targeted education programs.

In a discussion related to the feasibility of the green-roof, Stefano Cascone (2019) mentioned that there needs to be supported in the roles, requirements, characteristics, and materials, which are considered suitable for each green-roof layer [30]. The role needed in sustainable development includes the role of the community, in that, it is necessary to identify the role of the community in Indonesia to be able to prepare the strength for the implementation of green-roof innovations in a sustainable manner. The role of the community greatly influences the achievement of the success of the green concept towards energy efficiency and environmental sustainability. In a case study in the village of Gampingan-Pakuncen, Yogyakarta, through the research of Erni et al. (2019), the role of the community could be managed through controlling community activities in communal activities [31]. This indicates that the role of the community can also be an opportunity to rely upon the feasibility of green roofs.

*1.2. Research Aim*

This research aimed to find a relationship between the role of the community in green-roof feasibility, which is to get a more detailed picture of the role of the community from the aspect of awareness, knowledge, and participation in low, middle, and high-income communities; it also identifies correlations between residential locations and support for green-roof feasibility in Indonesia. The benefit of this research would be to obtain accurate and valid information in mapping Indonesian society to determine the patterns of interaction of development doers towards sustainable green-roof.

Several theories and research findings reveal the benefits of green-roof and the important role of the community in its sustainability, but very few studies discuss the role of Indonesian people in the feasibility of green-roof implementation. As such, the role of the community is not merely a complementary element of the development success, but also a key element that determines the development sustainability, which is the feasibility of the green-roof.

## 2. Materials and Methods

The research was a response networking of Indonesian people towards the implementation of green-roof in residential buildings. Community responses explored in the study included aspects of awareness, knowledge, and participation. The aspect of awareness is the level of response of the community in feeling the importance of playing an active role in efforts to strive for green-roofs in residential buildings. The knowledge aspect includes the community's insight into the benefits of green elements in dwellings, and how to provide green areas on residential roofs. Meanwhile, the aspect of community participation was obtained through questions in the form of questionnaires related to community interests, willingness to be involved in maintenance, and provision of green areas on residential roofs. The three aspects can contribute to illustrating the role of the community in green-roof feasibility.

*2.1. Location*

The study was conducted in Indonesia, including the island of Sumatra, i.e., the Samosir Toba area; on the island of Java, which includes Central Jakarta, Bekasi, Serang, Bandung, Yogyakarta, Kebumen, Semarang, Surakarta, Madiun, and Malang; and in East Nusa Tenggara, i.e., in Kupang. The areas represent regions with criteria for metropolitan cities, provincial capitals, cities, and developing cities. The basic consideration in selecting urban areas was the character of dense residential areas with a relatively high population.

## 2.2. Sampling Technique

The technique of distributing questionnaires was a random purposive that deals with one area or randomly adjacent. Such a technique took into account the diversity of regional profiles from the aspect of the visual arrangement of the environment in residential areas. Site selection included the area representing metropolitan, the provincial capital, urban, and rural area. The community, selected as respondents, was determined based on observations in the field, through consideration of criteria for low, middle, and high-income people. Data grouping included community responses, based on aspects of awareness, aspects of knowledge, and aspects of participation. Data grouping was also categorized as metropolitan, provincial-capital, urban, and rural communities. The results of the questionnaire collected were grouped and analyzed, statistically, using the chi-square method.

The research categorized the community for distributing questionnaires into three categories, i.e., low, middle, and high-income people. This categorization was intended to obtain information on whether there is a relationship between the economic status of the community and the support for green-roof sustainability.

The study accommodated 591 respondents, representing a variety of regions in Indonesia, including metropolitan, provincial capitals, urban, and rural. The distribution of questionnaires with various regional characteristics aimed to gain an overall picture of the response of the people who live in urban and rural areas. The distribution of the questionnaire was targeted at community representatives as a sample with 197 in each community, through field observations by the data-searching team, which was conducted from 2017 to 2018.

## 3. Results

### 3.1. Awareness Aspect

The aspect of public awareness is the level of concern for the importance of green-roof in residential buildings. The results of the analysis of the collected questionnaires are presented in Tables 1–3, showing the comparison of observations and analyses.

**Table 1.** Result of Observation from the Awareness Aspect.

| Criteria | Agree | Disagree | Amount |
|----------|-------|----------|--------|
| High-income community | 157 | 40 | 197 |
| Middle-income community | 149 | 48 | 197 |
| Low-income community | 151 | 46 | 197 |
| Total | 457 | 134 | 591 |

**Table 2.** Analysis of Expectation on the Awareness Aspect.

| Criteria | Agree | Disagree | Amount |
|----------|-------|----------|--------|
| High-income community | 152 | 45 | 197 |
| Middle-income community | 152 | 45 | 197 |
| Low-income community | 152 | 45 | 197 |
| Total | 457 | 134 | 591 |

**Table 3.** Value of Chi-square.

| Probabilities | 0.6054 |
|---------------|--------|
| Chi-Count | 1.8598 |
| Chi-Table | 0.1026 |

Analysis of the results from Table 3 determined that: HO = Public awareness of the importance of green-roof did not depend on the community income. H1 = Public awareness of the importance of green-roof depended on community income.

The decision rule stipulated that if Chi-Count > Chi-Table, then HO was rejected, and if Chi-Count < Chi-Table, then HO was accepted. There were two hypotheses, i.e., HO was that the public awareness of the green-roof practices was not influenced by economic level, and H1 was that the community participation towards green-roof practices was influenced by economic level. The results of the questionnaire indicated that Chi-Count > Chi-Table, meaning that HO was rejected with a small difference. The results indicated that the value Chi-Count 1.8598 > Chi-Table 0.1026, implying that the HO statement, about public awareness of the importance of green-roof does not depend on income, was rejected. Therefore, there was a relationship between community awareness of the importance of green-roofs and the level of community income. The higher the community income, the more it provided support for green-roof feasibility.

*3.2. Knowledge Aspect*

The knowledge aspects include basic knowledge about understanding green space on green-roofs, positive benefits, and opportunities for environmental sustainability. Table 4 is the result of the questionnaire calculation, which is distributed by 197 questionnaires in each classification of the community economic status, in that the total number of questionnaires was 591 respondents, which spread throughout Indonesia.

**Table 4.** Result of Observation from the Knowledge Aspect.

| Criteria | Agree | Disagree | Amount |
|---|---|---|---|
| High-income community | 170 | 27 | 197 |
| Middle-income community | 164 | 33 | 197 |
| Low-income community | 168 | 29 | 197 |
| Total | 502 | 89 | 591 |

The results of the data collection were processed using the chi-square statistical method in order to obtain a relationship between the economic level of the community and the carrying capacity of sustainability in the aspect of knowledge. The analysis in Table 5 results in an analysis of expectations, with the proportion of agreeing 167 and disagreeing of 30.

**Table 5.** Analysis of Expectation from the Knowledge Aspect.

| Criteria | Agree | Disagree | Amount |
|---|---|---|---|
| High-income community | 167 | 30 | 197 |
| Middle-income community | 167 | 30 | 197 |
| Low-income community | 167 | 30 | 197 |
| Total | 502 | 89 | 591 |

Analysis of the results in Table 6 stipulated that HO = community knowledge of the benefits of the green-roof did not depend on the community income, while H1 = community knowledge of the benefits of the green-roof depended on the community income. The decision rule was, if Chi-Count > Chi-Table, then HO was rejected; whereas, if Chi-Count < Chi-Table, then HO was accepted.

**Table 6.** Value of Chi-square.

| Probabilities | 0.6905 |
|---|---|
| Chi-Count | 2.3454 |
| Chi-Table | 0.1026 |

HO is public knowledge of green-roof practices and is not influenced by the economic level, and H1 is public knowledge of green-roof practices, which is influenced by the economic level. The results of the questionnaire showed that Chi-Count > Chi-Table, meaning HO was rejected by a very large difference, in that the conclusion was drawn that the role of community knowledge on the sustainability of the green-roof was strongly influenced by the level of the community's economy. The higher the income of the people with the economic status at the top level, they would have more extensive knowledge of the insight of the green-roof. However, the less the people's income, the less knowledge people would have of the existence of the green-roof innovations.

### 3.3. Participation Aspect

Table 7 describes the observation results of 591 respondents in several regions in Indonesia. The community responses to an active role in green-roof practice activities showed more than 50% support, but the percentage of respondents who did not support exceeded aspects of awareness and knowledge. A total of 71 respondents from low-income communities did not provide support, and this is the highest number compared to other strata of society. The results of the analysis in Table 8 obtained the proportion of 134, who agree, and 63, who disagree, at every level of society. The results of the analysis of participation aspects tended to have decreased support from the community compared to the aspects of awareness and the knowledge aspects.

**Table 7.** Result of Observation on the Participation Aspect.

| Criteria | Agree | Disagree | Amount |
|---|---|---|---|
| High-income community | 146 | 51 | 197 |
| Middle-income community | 129 | 68 | 197 |
| Low-income community | 124 | 71 | 197 |
| Total | 401 | 51 | 591 |

**Table 8.** Analysis of Expectation on the Participation Aspect.

| Criteria | Agree | Disagree | Amount |
|---|---|---|---|
| High-income community | 134 | 63 | 197 |
| Middle-income community | 134 | 63 | 197 |
| Low-income community | 134 | 63 | 197 |
| Total | 401 | 190 | 591 |

Analysis of the results in Table 9 stipulated that HO = Community participation in the feasibility of green-roof did not depend on community income, while H1 = Community participation in the sustainability of green-roof depended on community income. The decision rule stipulated that if Chi-Count > Chi-Table, then HO was rejected, whereas if Chi-Count < Chi-Table, then HO was accepted.

**Table 9.** Value of Chi-square.

| Probabilities | 0.0586 |
|---|---|
| Chi Count | 0.1208 |
| Chi Table | 0.1026 |

The research hypothesis consisted of two choices i.e., HO = Community participation in green-roof practices was not influenced by the economic level, and H1 = the economic level influenced community participation in green-roof practices. The results of the questionnaire showed that Chi-Count > Chi-Table, meaning HO was rejected with a small difference, in that the conclusion drawn was that the role of community participation in the sustainability of green-roof was influenced by the economic level.

In another part of the study that compiled the results of some data, it was indicated that community responses to the existence of a green-roof tended to give support, as described in Figures 2–5. Figure 2 presents the proportion of people who support and do not support green-roof innovation, from the community of low, middle, and high-income.

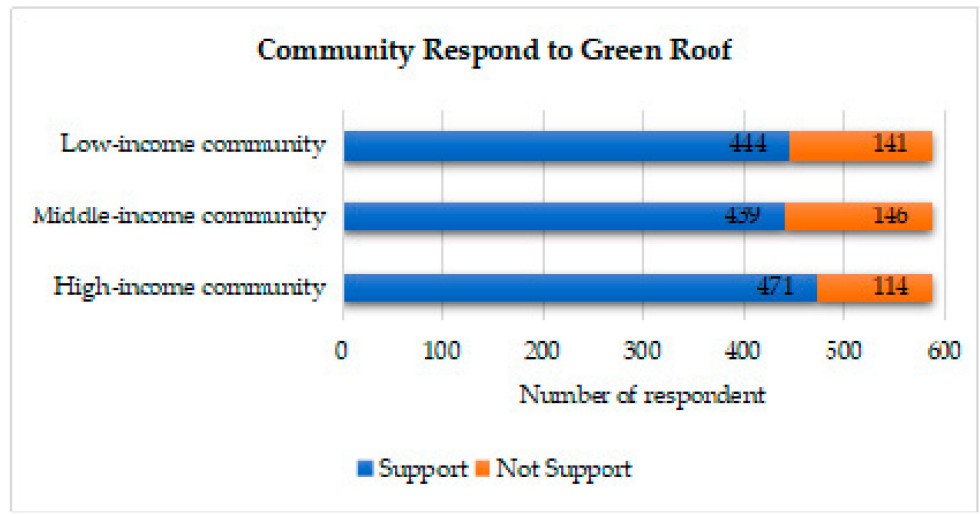

**Figure 2.** Comparison of support for green-roof based on the economic level of the community.

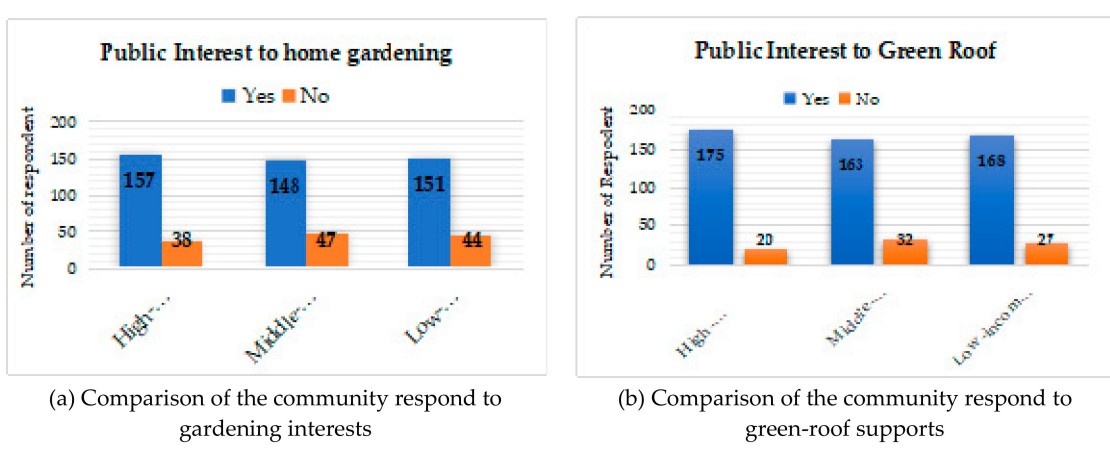

(a) Comparison of the community respond to gardening interests

(b) Comparison of the community respond to green-roof supports

**Figure 3.** Comparison of interests in (**a**) gardening activities and (**b**) green-roof based on the economic level of the community.

Figures 3–5 present a smaller amount of data by taking representative samples from all walks of life, ranging from low, middle, and high-income people. Figure 4 is a statistic of the community interest in gardening activities in residential areas and gardening activities on the green-roof. The results showed that most of the community gave support to gardening activities in their respective places of residence, especially for gardening activities in the green-roof area that has a high-interest rate. Whereas, Figure 4 indicates the gardening capabilities of the Indonesian people with the results of more than 60% from all walks of life having gardening abilities. The more valuable aspect of participation in the sustainability of green-roof was the willingness to prepare green spaces in the roof-area, as shown in Figure 5. The results of the study indicated somewhat encouraging numbers, although the number that did not support was actually more than other aspects.

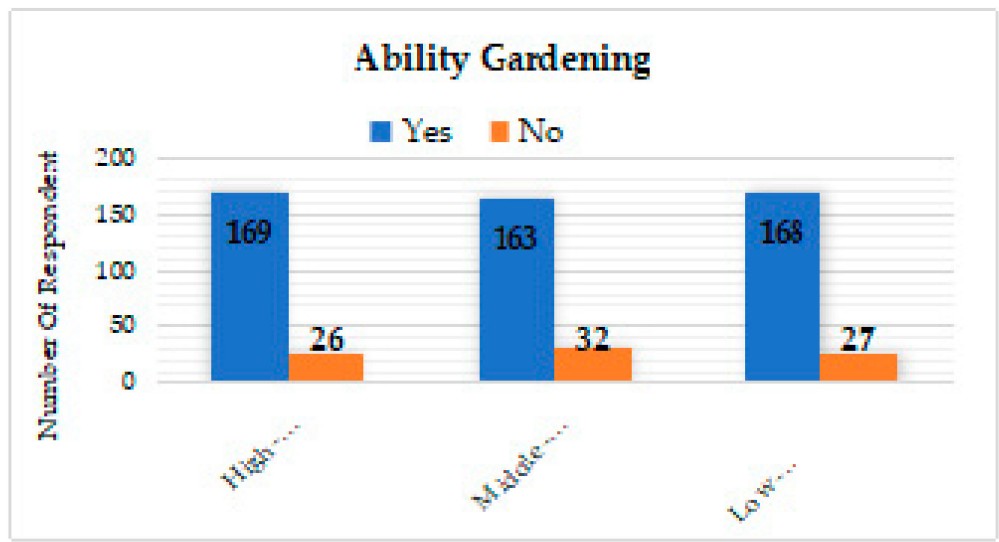

**Figure 4.** Comparison of gardening capabilities based on the economic level of the community.

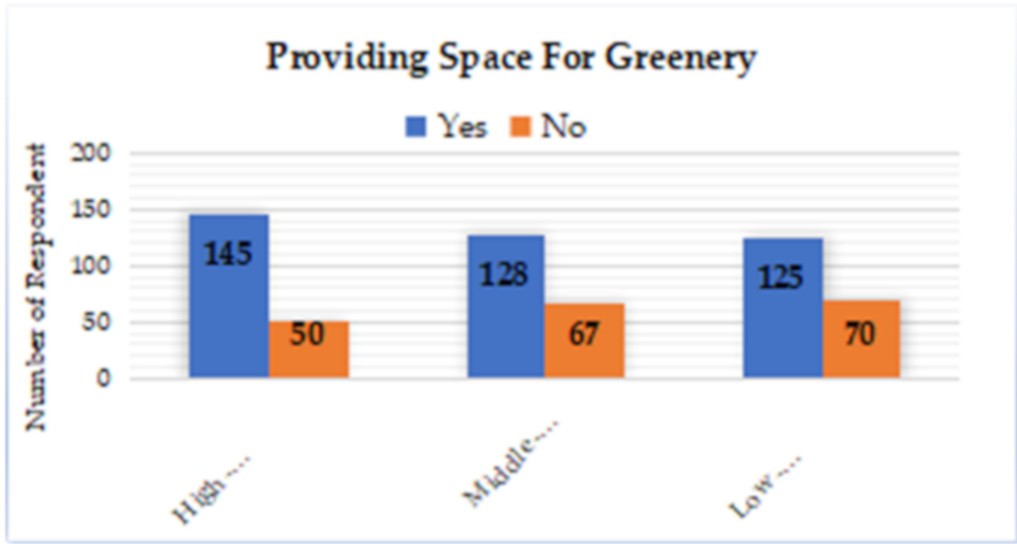

**Figure 5.** Comparison of the ability to provide gardening areas based on the economic level of the community.

The results of the community responses based on residential location were obtained, as shown in Figures 6–8. Figure 6 shows a comparison of responses of community members that resided in metropolitan areas, where the knowledge level of the importance of green-roofs relatively supported the existence of green-roof in residential buildings. People who lived in metropolitan cities in Indonesia apparently had a less supportive response to an active role in participating in the existence of green-roof. In the aspect of participation, middle and low-income people tended to disagree with the provision of green-roof for occupancy; whereas, the aspects of awareness and knowledge had high supportive responses.

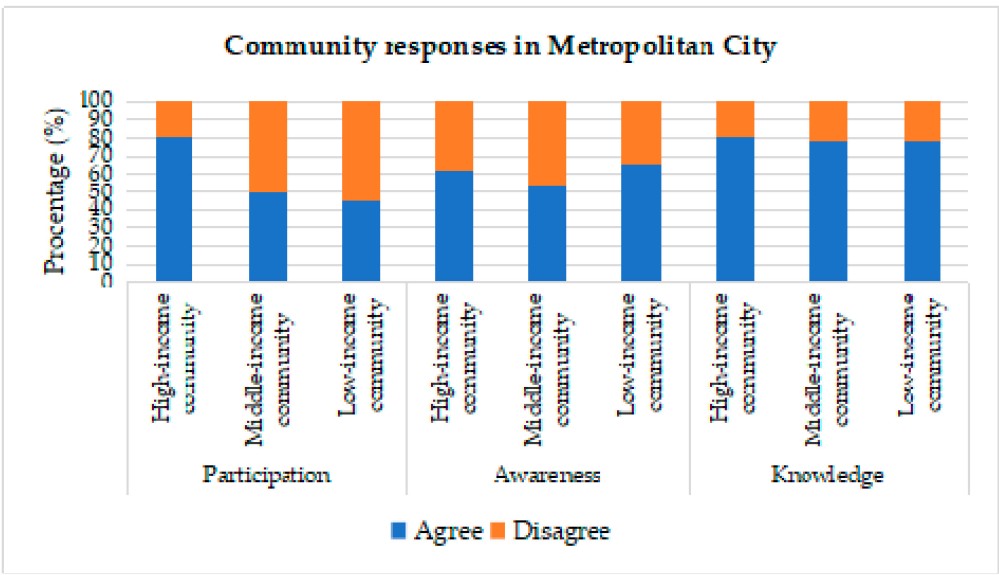

**Figure 6.** Comparison of community response to green-roof in the Metropolitan.

The response of people living in the provincial-capital region had a different character from the people living in metropolitan cities. In general, people from low to high-income levels provided support of more than 50%, and people who had not yet supported was more than 30%, as shown in Figure 7. Meanwhile, for the aspect of awareness and knowledge, people living in the provincial-capital region had more support than the aspect of participation.

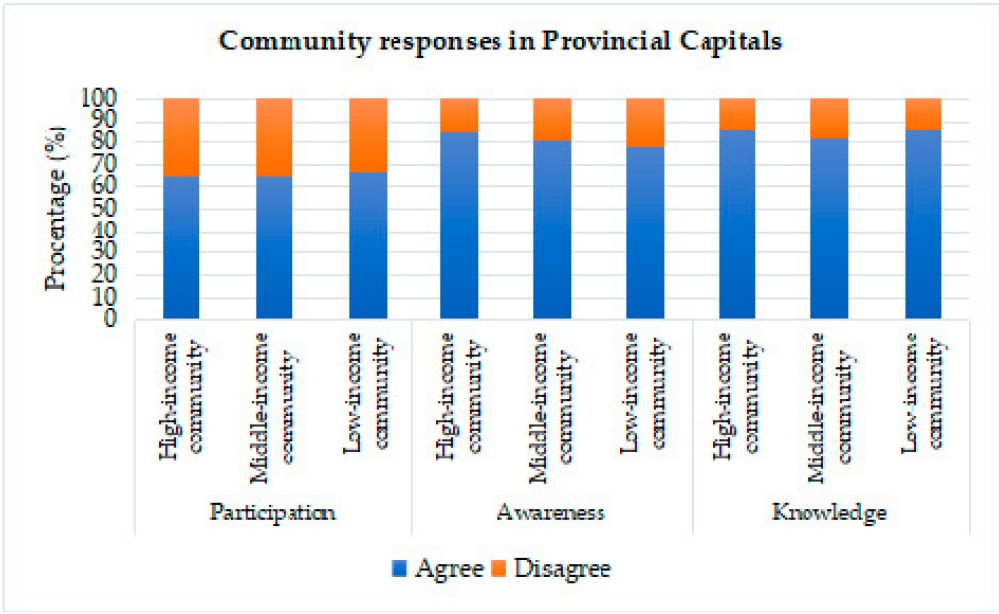

**Figure 7.** Comparison of community response to green-roof in the provincial capitals.

The response of people living in urban areas is illustrated through the graph in Figure 8. Overall, there were still views of urban communities that had not provided support for the existence of green-roof in residential areas. People from low, middle, and high-income groups showed less support in the aspect of participation. Specifically, for low-income people in urban areas, the number of support was less than optimal, both in the aspects of participation, awareness, and knowledge.

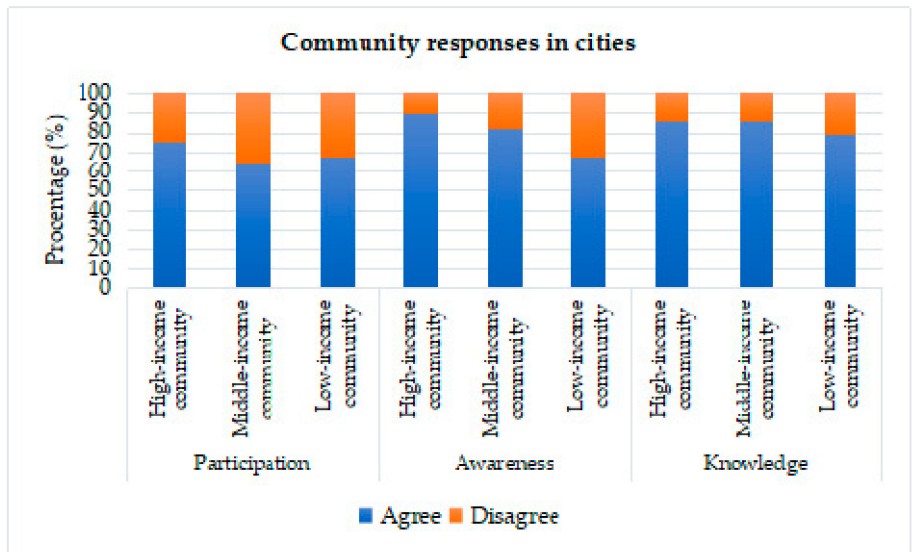

**Figure 8.** Comparison of community response to green-roof in cities.

People in rural areas considered green-roof to be less useful, especially in low-income communities. The low-income community response was significantly less supportive in terms of an active role to participate in providing area on the roof of their dwellings, as shown in Figure 9. Low-income people felt that green-roof innovation in dwellings was neither necessary nor urgent. Another aspect that lacked support from low-income people in rural areas was the level of awareness that is still not widespread in the importance of green-roof.

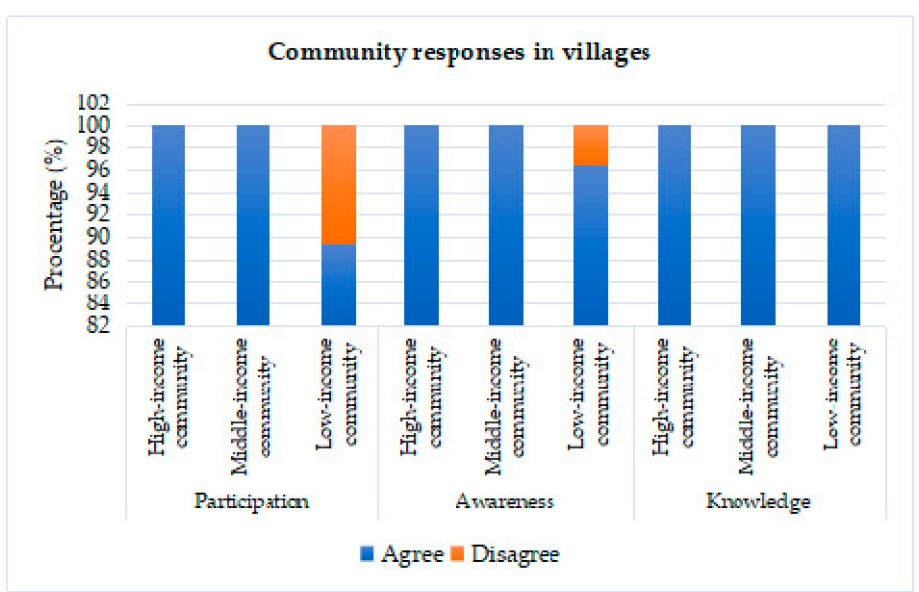

**Figure 9.** Comparison of community response to green-roof in the villages.

## 4. Discussion

According to Tricarico, the main way to accommodate the role of the community in sustainable development includes four things, including knowing the limits of community values, promoting equal democracy in society, neutral communication, and understanding the level of the community knowledge [28]. Community value limits are social values that are formed from society, which naturally adjusts the culture of the community. In people's lives that already have social value, there are areas of individual, family, and environment. The community values the environment as part of the space

used communally. This theory was further developed by green-roof research that focused on the prospect of sustainability through the role of the community. The results of the study, comparing the effect of the relationship between the economic level of the community with the support of the role of green-roof feasibility, indicated that there was a close relationship between economic capacity and the contribution of the community's role to the sustainability of green-roof. The results of the analysis on the relevance of the Indonesian people's economic level to the sustainability of the concept of green-roof for occupancy, which is based on aspects of participation, knowledge, and awareness, on average, indicated very relevant results. The results of the study in Tables 1–9 indicated the closest relationship between the level of income and carrying capacity of the awareness aspect in Table 6, obtained a Chi-Count of 2.3454 and a Chi-Table of 0.1026, with a very large difference. The results of the study tended to illustrate that the sustainability of green-roof was highly dependent on the role of the community. Furthermore, the results obtained with high-income people could be a mobilizer to facilitate low or middle-income communities. The economic strength of the community provided support for the implementation of green-roof, but the support of low and middle-income communities needed to be involved in order to create mutually beneficial cooperation. Thus, this study corroborated previous research [13,25,26].

Previous research by Sri Yuliani [29] that discussed the role of the community towards the sustainability of green areas in Surakarta concluded that it was important to involve the role of the community in the implementation of green spaces that support sustainable development. The role of the community contributes to the success of the sustainability of the green design provided in buildings, especially in cities in areas with humid tropical climates, such as Indonesia, indicating that the largest land use is housing area [29]. The research was explained in more detail in the analysis of the calculation results obtained, that the high economic level community had a higher percentage in supporting the sustainability of green concept practices, as shown in Figure 2. This was due to the capital owned by high-income people could be stronger in supporting the ability to hold green-roof installation.

Research that raises the role of the community towards the sustainability of green-roof is an attempt to get a detailed picture of previous research on the role of the community in the sustainability of green design. The results of previous studies concluded that the role of the community is very strategic in supporting the sustainability of green design. The role of the community is categorized into three levels, including low, middle, and high-income people. Categorizing respondents in this study was to find out which elements of the community had strategic opportunities for green-roof sustainability. In general, the results presented in Tables 1–9, in this study, revealed that there was a relationship between the economic status of the community and the carrying capacity of the green-roof sustainability, on the aspects of awareness, knowledge, and participation. Meanwhile, among the three aspects needed to support the sustainability of green-roof, the participation aspect was the lowest available in Indonesian society. Based on the findings of research on aspects of participation, this study reinforced the findings of previous research conducted by Shin [9], including the need for economic power in green-roof installations. The provision of green roofs that require costs exceeding conventional roofs indirectly affects the readiness of the community in organizing green-roof on occupancy. As for the aspect of awareness, it tends to get higher support when compared to the aspect of participation, even though only on a small difference in numbers. The knowledge aspect is a huge potential possessed by the Indonesian people, where the community has a high ability to manage green areas. This potential needs to be considered to build the sustainability of green-roof as part of the expansion of residential roofs that have the potential for productive activities.

In detail, Figures 2 and 3 are visualizations of the results of data analysis on detailed aspects of awareness, where the percentage of carrying capacity of the community was very large and occurred in all three levels of society, i.e., low, middle, and high-income earners. Likewise, in Figure 4, it indicated a high number of supports in the aspect of knowledge, where this indicated that the public had adequate knowledge in the management of green-roof. However, in the aspect of participation, especially the willingness to hold a green-roof in each dwelling, it looked less enthusiastic than the

other two aspects, although it was still dominated by the support that reached more than 50% of the supporting community. Reduced support was due to the limitations of middle and low-income people who had minimal residential land. The results of this study indicated significant findings by Eunha Shin and Heungsoon Kim [9] that from an economic aspect, indeed, a green-roof required more cost than a conventional roof. Thus, the role of high-income community participation would be more supportive than middle-income or even low-income communities. However, the results presented in Figure 5 indicated that the ability to provide green space could be anticipated with high ability in gardening activities, as shown in Figure 4, through space efficiency and roof expansion as a productive green space. Therefore, it is necessary to have a pattern of cooperation that overlaps between levels of society.

The other part of the study indicated that there were different characters between people who live in metropolitan cities, provincial capitals, urban areas, and rural areas. For people who live in high densities, such as metropolitan areas, they had a tendency to provide lower support than rural communities. Different patterns of support could be caused by different demands on living between the areas. Tight competition in the survival for people who live in urban areas drained time and energy in that there were not enough roles that could contribute to the sustainability of green-roof.

High-income people who live in the rural area provided the highest support compared to others. While low-income people who live in a rural area, indeed, tended to provide less support for the green-roof. This could be understood because the area of green land in the rural area is relatively more available than in the urban area.

## 5. Conclusions

The study concluded that the role of the community in the sustainability of green-roof is very necessary because the community can determine the sustainability of the function of the green-roof productively. This is evidenced by the presence of abandoned green area facilities due to the low role of the community in participating in caring-for and maintaining. The most important role of the community in green-roof sustainability is awareness, knowledge, and active participation in maintaining and caring for the green-roof. Meanwhile, the role of the community in providing green areas on the roof of the building will provide productive value to the buildings occupied, especially when the green-roof is planted with productive plants. Placement of green-roof on residential buildings is very strategic because of the function and ease of access to carry out activities in productive green areas in the occupied housing.

In general, low, middle, and high-income people tend to have the opportunity to provide support for the sustainability of green-roof, even at different levels. The research concludes that there is a very significant relationship of the role of the community with differences in economic status in support of green-roof sustainability. Therefore, it needs a pattern of mutually beneficial interactions in the implementation of green-roof in residential.

The characteristics of low-income to high-income people in dense residential areas tend to neglect the importance of green-roof, even though the densely populated urban areas have limited green areas and limited residential land. Whereas rural communities with various layers of the economy, low, middle, and high-income have more support. The role of people who live in dense areas, such as metropolitan cities, provincial-capitals, and cities, needs to be increased in stages through aspects of awareness, knowledge, and participation. The role of the community can be managed by creating a gardening community network, which contributes to each other according to the economic capacity of the community. In worldwide, the gardening community network could be useful in other similar typical countries toward sustainable development.

The data of this research do not either consider the specific climatic region and kinds of roof material, but focus on the characteristic of community based on the economy-level to support the green-roof application. This research recommends the need for further research relating to the management and community empowerment strategies that play an active role in green-roof

sustainability. Further research should help explore more issues in the innovation field of green-roof implementation in Indonesia.

**Author Contributions:** Conceptualization, S.Y.; methodology, S.Y, and E.S.; validation, G.H. and E.S.; formal analysis, S.Y., G.H., and E.S.; resources, S.Y.; data curation, S.Y.; writing—original draft preparation, S.Y.; writing—review and editing, E.S.; visualization, S.Y.; supervision, G.H.; project administration, S.Y.; funding acquisition, S.Y. All authors have read and agreed to the published version of the manuscript.

**Funding:** This research was funded by The Research and Community Service Institution of Universitas Sebelas Maret (LPPM UNS), which has supported the research funding through the PNBP UNS.

**Acknowledgments:** The authors would like to give gratitude to the Doctoral Program of Architecture Sciences and Urban, Department of Architecture, Faculty of Engineering, Universitas Diponegoro, on supporting this research. We also give our thanks to the respondents and questioner collectors.

**Conflicts of Interest:** The authors declare no conflict of interest.

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
