# Peer review of "Green-Roof: The Role of Community in the Substitution of Green-Space toward Sustainable Development"

_sustainability, doi:10.3390/su12041429_

Round 1
Reviewer 1 Report
Major comments and suggestions for authors
The manuscript subject is interesting. However, the results and analysis are not very clear. More specific comments are as follows:
1- The analysis in the study is about the feasibility of green roofs in different communities, not sustainability.
Sustainability shows the ability to be maintained at a certain rate or level or the ability to exist continually. Feasibility shows that a project/idea is legally, economically, and technically practical and possible.
2- Line 14: “the role of the community for food security based housing sustainability.” In which part of the study, “food security” has been analyzed?
3- Line 16: “Data were analyzed using relevance statistics.” Which type of statistical analysis has been done? Most of the graphs and tables are just based on the percentage.
4- Line 20: “Therefore, it is necessary to educate green-roof technology to the community.” In all of the tables and graphs, most of the response was “agree” and “yes.” So why more education is necessary?
5- Line 94: it seems that the feasibility of green roofs in different communities has been analyzed, not sustainability so that the aim can be corrected.
6- “Location” and “Sampling Technique” can be in “Materials and Methods” not in “result.”
7- Sampling Technique:
The base for low, middle, and high-income people in each community of metropolitan, provincial-capital, urban, and rural communities is not clear in the paper. How many questionnaires have been collected in each community?
8- Since the green roof is counting as an LID (low impact development) method, in which sense the analysis of green roof has been done for rural areas?
9- The data of the selected locations such as the roof type, density, population, climate type, precipitation, average temperature not exist in the study. In this way, the results can not be used for other locations.
Author Response
I have revised the manuscript.

Reviewer 2 Report
The paper deals with the benefits of green-roof and the important role of the community in its sustainability.
The topic is worth of interest and centered on the journal scope. The paper is well structured, I suggest only minor revions .
In particular 'Abstract', in the present form, is not so clear, it should be rearranged following the structure of the Sections of the paper.
In 'Introduction' a new sub-Section 'Research aim' could be added.
In 'Conclusions', the authors should explain how the results found for Indonesia could be useful in worldwide.
The 'References' could be extended, for example citing at least:
a) DOI=10.1016/j.jobe.2018.11.009 (Journal of Building Engineering, ELSEVIER, 2019), togheter with previous Refs 6-8;
b) DOI=10.17660/ActaHortic.2018.1215.15 (Acta Horticulturae, ISHS, 2018), togheter with previous Refs 20-21.
In 'Results', in particular in Figures 6-7-8, the percentages should be expressed by integer numbers without decimals (as the authirs did in Figure 9).
Author Response
I have revised the manuscript.

Round 2
Reviewer 1 Report
No new comments.